# Fast semantic segmentation method for machine vision inspection based on a fewer-parameters atrous convolution neural network

**Jian Huang, Liu Guixiong\*, Binyuan He**

School of Mechanical and Automotive Engineering, South China University of Technology, Guangzhou, Guangdong Province, China

\* megxliu@scut.edu.cn

**Data Availability Statement:** The PASCAL VOC Dataset used to support the findings of this study is available at http://host.robots.ox.ac.uk/pascal/VOC/. The CITYSCAPES Dataset used to support

## Abstract

Owing to the recent development in deep learning, machine vision has been widely used in intelligent manufacturing equipment in multiple fields, including precision-manufacturing production lines and online product-quality inspection. This study aims at online Machine Vision Inspection, focusing on the method of online semantic segmentation under complex backgrounds. First, the fewer-parameters optimization of the atrous convolution architecture is studied. Atrous spatial pyramid pooling (ASPP) and residual network (ResNet) are selected as the basic architectures of $\eta_{\text{seg}}$ and $\eta_{\text{main}}$, respectively, which indicate that the improved proportion of the participating input image feature is beneficial for improving the accuracy of feature extraction during the change of the number and dimension of feature maps. Second, this study proposes five modified ResNet residual building blocks, with the main path having a 3 × 3 convolution layer, 2 × 2 skip path, and pooling layer with $l_{\text{s}} = 2$, which can improve the use of image features. Finally, the simulation experiments show that our modified structure can significantly decrease segmentation time $T_{\text{seg}}$ from 719 to 296 ms (decreased by 58.8%), with only a slight decrease in the intersection-over-union from 86.7% to 86.6%. The applicability of the proposed machine vision method was verified through the segmentation recognition of the China Yuan (CNY) for the 2019 version. Compared with the conventional method, the proposed model of semantic segmentation visual detection effectively reduces the detection time while ensuring the detection accuracy and has a significant effect of fewer-parameters optimization. This slows for the possibility of neural network detection on mobile terminals.

## Introduction

Semantic segmentation is a basic task in computer vision and is aimed at dividing a visual input into different semantically interpretable categories (that is, assigning a semantic label to each pixel of the image [1]). Although unsupervised methods, such as clustering, can be used

the findings of this study is available at http://cityscapes-dataset.com. The DeepLab v3 pretrained model are the implement of GLUON-CV, and available at https://gluon-cv.mxnet.io/model_zoo/segmentation.html. We provide the DeepLab v3 with modified ResNet50, and a program (*.ipynb) to use it on different image of PASCAL VOC, CITYSCAPES or CNY. https://github.com/HJ0116/utilization_semantic_segmentation

**Funding:** This work was supported in part by the Key-Area Research and Development Program of Guangdong Province, China under Grant 2019B010154003, and in part by the Guangzhou Science and Technology Plan Project under Grant 201802030006.

**Competing interests:** The authors have declared that no competing interests exist.

for segmentation, their results do not necessarily possess semantics. Fig 1 presents the flow-chart of the general method of semantic segmentation, including input, area generation, feature extraction, classifier, post-processing, and output segmentation results. The flow-segmentation method is composed of multiple independent algorithm modules, which are designed by experts; for example, the second-order pooling (SOP) [2], discriminative re-ranking of diverse segmentations (DRS) [3], unified detection and segmentation (UDS) [4] and simultaneous detection segmentation (SDS) [5].

Semantic segmentation has the best fine-grained identification and can distinguish the entity levels/backgrounds (centroid) of the components through indirect calculation. The method also has a clear physical definition and is suitable for machine-vision identification with high spatial resolution and reliability. The general machine-vision system belongs to a complex photoelectric system with high requirements of accuracy, real-time applicability, and repeatability [6–9]. A convolutional neural network (CNN) for semantic segmentation was applied to assign a semantic label to each pixel [10]. The CNN semantic segmentation method is derived from a single-step end-to-end CNN semantic segmentation model divided into multiple modules for avoiding extensive processing. However, the connection mode of various modules directly affects CNN, while rendering of the model increases dramatically. The required vast storage and computational overhead severely limit the application of CNN in the field of low power consumption for visual detection.

Atrous convolution was first presented in the field of image segmentation. In this method, the image is input into the network to extract features through CNN. Subsequently, the image scale is reduced by pooling while the receptive field is increased. Although the CNN semantic segmentation method is a single-step end-to-end semantic segmentation, it is not divided into multiple modules. Nevertheless, the connection mode of various modules directly affects CNN.

Owing to the rapid development in hardware and computing power, the fewer-parameters network structure can effectively solve the problem of training and prediction efficiency due to the modeling complexity. The prediction efficiency is mainly based on the model storage and prediction speed. The storage problem is mainly due to the presence of hundreds of networks, which in turn indicates dealing with a great number of weighting parameters; the storage of a large number of weighting parameters demands for a large device memory. Moreover, the speed problem is mainly based on practical applications. The atrous convolution architecture eliminates part of the CNN pooling layer while replacing the convolutional layer with a cascade or parallel atrous convolutional layer, enabling the analysis of the feature map at multiple arbitrary scales, and thus significantly improving the segmentation accuracy [11–13] and providing the possibility of detecting applications in the field of low power consumption. For obtaining a more accurate and faster fewer-parameters model as well as a method to achieve online machine vision and identification, in this study, the weight optimization technology

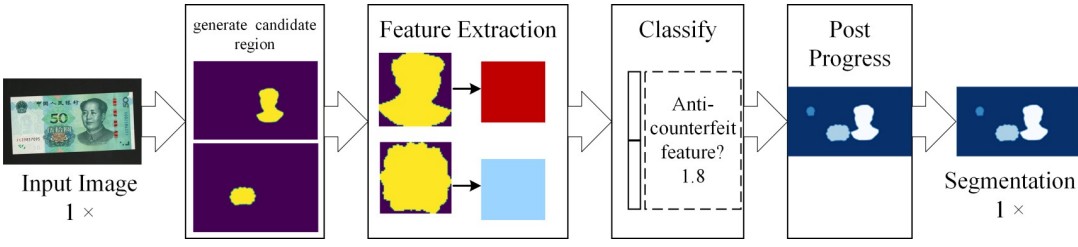

**Fig 1. Flowchart of the general semantic segmentation method.**

was investigated based on the downsampling atrous convolution architecture-network model. This paper first presents a discussion of the associated indicators, including accuracy, computing speed, space occupation, and training speed, and then the modeling of the fewer-parameters of atrous convolutional architecture, by selecting $\eta_{seg}$ and $\eta_{main}$ as the basic architecture. Next, an analysis of the fewer-parameters mechanism of downsampling residual blocks to improve the residual network (ResNet) by using $d_{main}$ and $d_{main}$ is presented to show that a ResNet structure can be modified by downsampling residual blocks.

The main contributions of this article are as follows:

1. Based on the parameters of the dense convolutional architecture network, including dense prediction network structure $\eta_{seg}$, main network structure $\eta_{main}$, and main network depth $d_{main}$, we designed a fewer-parameters optimization mathematical model according to the atrous convolution architecture network.

2. We analyze ResNet downsampling residual construction blocks and point out that improving input utilization during downsampling is beneficial for improving the accuracy of the ResNet. The fewer-parameters optimization of ResNet and $d_{main}$ could be accomplished based on the downsampling of residual blocks.

3. We adopted the downsampled residual block to improve the ResNet structure after comparing the modified structure with the $3 \times 3$ main path and $l_s = 2$ convolutional layers; $2 \times 2$ skip path and $l_s = 2$ pooling layers; $3 \times 3$ main path, $2 \times 2$ skip path, and $l_s = 2$ pooling layers. The application of machine-vision segmentation in the 2019 version of the China Yuan (CNY) showed that the proposed modified network benefits from the increase of the mean intersection over union ($I\bar{o}U$) and the decrease in the segmentation time, $T_{seg}$.

The remainder of this article is organized as follows. Section 2 presents the recent studies related to our study, including a multi-empty convolutional network structure and RestNet representative network. The proposed method is presented in Section 3, with the focus on three aspects: 1) a discussion related to evaluation indicators; 2) the fewer-parameters optimization modeling and selection of the basic architecture of $\eta_{seg}$ and $\eta_{main}$; and 3) the improved fewer-parameters mechanism of ResNet and $d_{main}$ based on the downsampling of residual blocks as well as the modification of the ResNet structure based on this downsampling. The experimental application of the proposed method is presented in Section 4 and the conclusions are drawn in Section 5.

## Related work

The current CNN end-to-end semantic segmentation method can process images of any resolution, identify multiple objects at the pixel scale, and output multivalue maps without resolution loss [14]. This process can be integrated into machine vision, thus forming an artificial intelligence method with great generalization in precision measurement and analysis. Unlike the codec architecture that retains the pooling layer as the encoder, the atrous convolution architecture establishes other techniques to achieve semantic segmentation. Atrous convolution removes part of the pooling layer and replaces it with the convolutional layer and the fully connected layers with the atrous convolution to maintain a high resolution of the feature map. ResNet is the ILSVRC 2015 champion network [15]. The main idea of the residual block structure is to increase the direct connection channel in the network; that is, the idea of the highway network, which effectively solves the problem of numerous network layers, causing gradient dispersion or gradient explosion problems. We analyzed the research progress on the atrous

convolution and ResNet network structure to provide the feasibility and theoretical basis for subsequent research.

## CNN-based semantic segmentation

Typical CNN-based semantic segmentation networks include fully convolutional networks (FCN) [16], SegNet [17], and pyramid scene parsing network (PSPNet) [18].

FCN [16] is an end-to-end semantic segmentation network proposed by Jonathan Long et al. in 2014. Its distinguishing characteristic is the conversion of a fully connected layer into a convolutional layer. It is capable of processing images of any resolution, successfully overcoming the limitation of a fully connected layer—that is, only being capable of processing images of a specific resolution. However, FCN suffers from certain problems, including the loss of details, smoothening of complex boundaries, and poor detection sensitivity in the case of small object. FCN gain a mIoU accuracy 62.2% on PASCAL VOC 2012. SegNet [17] is an efficient, real-time, and end-to-end semantic segmentation network proposed by Alex Kendall et al. in 2015. It identifies a one-to-one correspondence between the decoder and encoder by using the maximum pooling index of the encoder to perform non-linear upsampling to form a sparse feature image. Following this, a dense feature map is generated via convolution. However, the accuracy of SegNet is lower than that of FCN, and it also suffers from the issue of boundary misdetection. SegNet gain a mIoU accuracy 59.9% on PASCAL VOC 2012. PSPNet [18] was proposed by Zhao Hengshuang in 2017, PSPNet exploiting the capability of global context information by different-region-based context aggregation through a novel pyramid pooling module together with the proposed pyramid scene parsing network. PSPNet's global prior representation is effective to produce good quality results on the scene parsing task, while PSPNet provides a superior framework for pixel-level prediction tasks. A single PSPNet yields mIoU accuracy 85.4% on PASCAL VOC 2012.

## Atrous CNN-based semantic segmentation

Typical atrous CNN-based semantic segmentation networks include DeepLab [19] and dilated convolution network (DCN) [20], DeepLab v2 [21] and DeepLab v3 [22].

DeepLab [19] was proposed by the University of California and Google in 2015. Fig 2 shows the structure of the DeepLab model, in which the last two pooling layers of CNN are removed and $\alpha$ = 2 and 4 atrous convolution layers are used to replace a convolutional layer and a fully connected layer, respectively. Thus, the network outputs a 1/8 original-resolution

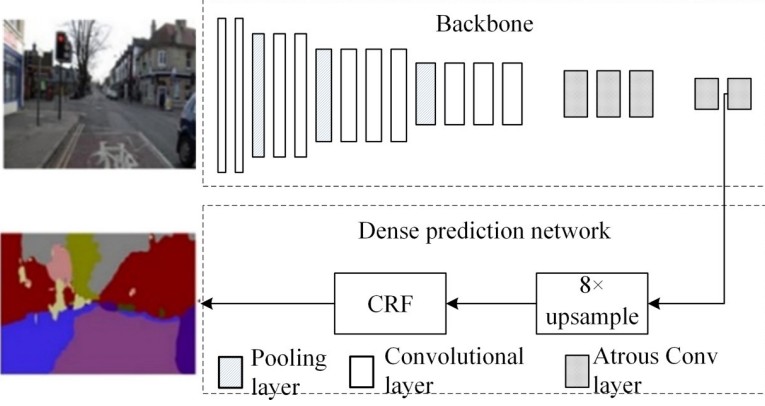

**Fig 2. DeepLab model structure diagram.**

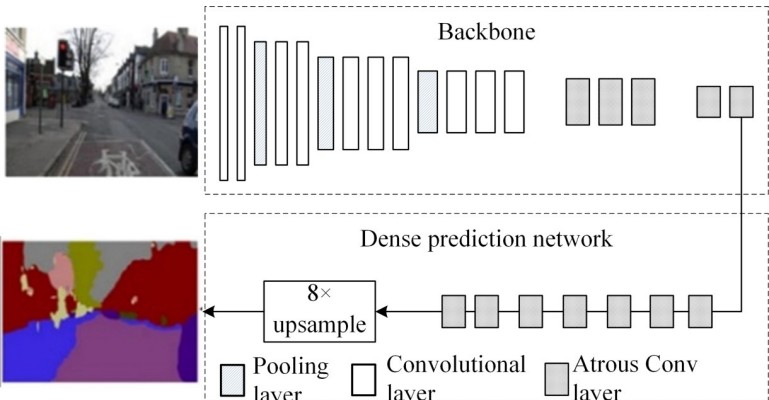

**Fig 3. DCN model structure diagram.**

feature map (which is more than that of the feature map outputted by FCN) and then upsamples it to restore the original resolution by using a conditional random field to improve boundary prediction accuracy [23]. DeepLab proposes a fundamental paradigm of cascaded atrous convolution architecture. The segmentation of a 480 × 360 image by using DeepLab requires 110.06 ms, which is 73.95% and 65.29% faster than the SegNet and FCN methods of the codec architecture. Moreover, its $I\bar{o}U$ was tested at 67.1% for VOC2012.

In 2016, Princeton University and Intel Corporation proposed DCN [20]. The atrous algorithm includes signal decomposition at multiple scales, and atrous convolution is applied at multiple scales to extract feature maps. Fig 3 depicts the structure of the DCN model, comprising a set of cascaded context networks with seven 3 × 3 atrous convolutions and one 1 × 1 convolution based on DeepLab; the model uses multiscale information to improve accuracy. The DCN has an $I\bar{o}U$ of 67.6% in VOC2012, 0.5% higher than DeepLab.

In 2016, the University of London and Google proposed DeepLab v2 [21]. Fig 4 shows the model of the DeepLab v2 structure using the atrous spatial pyramid pooling (ASPP) instead of the fully connected layer in CNN. ASPP is composed of four atrous convolutions with atrous ratios of 6, 12, 18, and 24, used in VGG and ResNet obtained an $I\bar{o}U$ of 71.6% and 79.7%, respectively, in VOC 2012. DeepLab v2 can consistently detect large object areas.

In 2017, Google proposed DeepLab v3 [22], as illustrated in Fig 5. This model improved the ASPP module by introducing a 1 × 1 convolutional layer and global pooling layer for maintaining features. It modifies the feature fusion method that stiches the output feature map in the third dimension, and then upsamples it to restore the resolution. DeepLab v3 could obtain an $I\bar{o}U$ of 86.9% in the VOC2012 dataset.

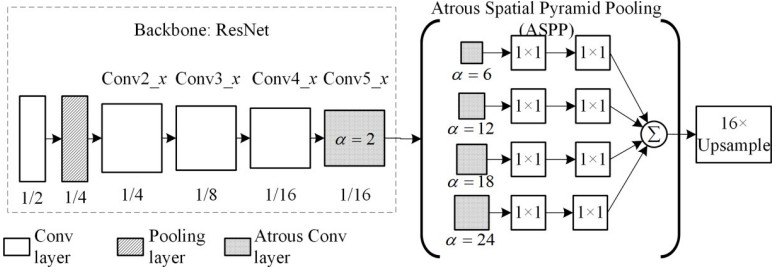

**Fig 4. Model of DeepLab v2 structure.**

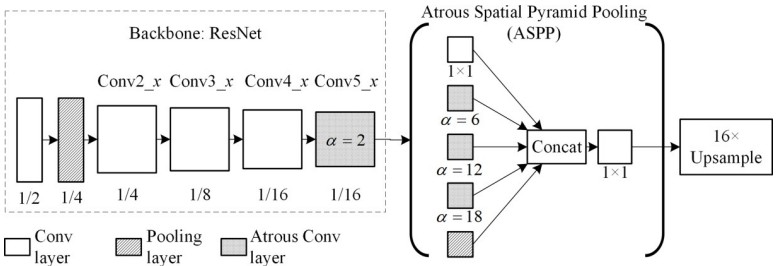

**Fig 5. Model of DeepLab v3 structure.**

## ResNet network

He et al. [15] proposed ResNet, in which the "degeneration" problem is solved through the residual block model. The main idea of the residual block structure in ResNet is to add the direct connection channel into the network, i.e., the idea of a highway network. Fig 6 shows the structure of the ResNet network. After the image is input, the short connection of the dimension matching converts from a dotted line into a solid line. When the dimensions do not match, two equivalent mapping methods can be selected, i.e., directly adding zero to increase the dimension or multiplying the W matrix to project into a new space. This allows the network to theoretically always remain in an optimal state, and thus the performance will not deteriorate with depth.

When the model becomes more complicated, some problems could arise; for instance, the accuracy could drop rapidly after saturation, resulting in higher training errors or the

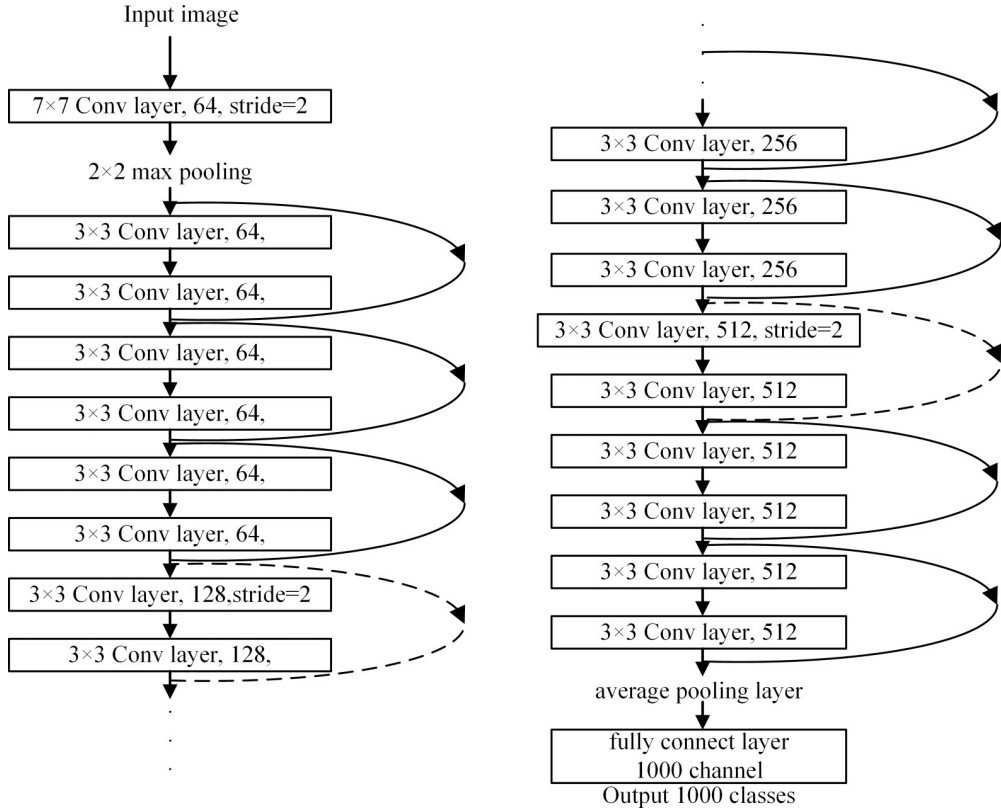

**Fig 6. Structural model of ResNet.**

stochastic gradient descent (SGD) optimization could become more difficult. To solve these problems, the residual structure is used such that the depth of the network model is unlimited in a large range (currently more than 1000 layers). The error rate of ResNet on Top-5 is 4.49%, and the number of network model parameters is less than those used in VGGNet with outstanding results. ResNet is representative of the current deepening model but its deepening increases the network size, and thus more storage space is required.

Our preliminary research [24] found that, replacing the FCN's VGG16 backbone with ResNet50, $I\bar{o}U$ on VOC2012 increase from 62.6% to 83.6%.Yu et al. [25] replacing the DeepLab's VGG16 backbone with ResNet50, $I\bar{o}U$ on VOC2012 increase from 67.1% to 75.6%. Lu et al. [26] replacing the DCN 's VGG backbone with ResNet50, $I\bar{o}U$ on VOC2012 increase from 67.6% to 81.0%.

The atrous convolutional network has a greater advantage in improving the accuracy of boundary prediction, and it has improved the performance of semantic segmentation in recent years. Currently, ResNet network has the advantages of high accuracy in classification and an excellent $I\bar{o}U$ combined with multiple semantic segmentation networks. Therefore, further research on the fewer-parameters optimization of the model of the atrous convolution network architecture can fulfill the requirements of accuracy and its speed in the online machine vision detection and identification.

## Proposed method

The proposed method considers the pixel accuracy *PA* and m*IoU* $I\bar{o}U$ [27]. Segmentation time $T_{\text{seg}}$ is defined as the time the algorithm requires to segment the image, and $I\bar{o}U$ is defined as the degree of overlap between the segmentation result and its true value. Under the condition that $I\bar{o}U$ satisfies the minimum $IoU_{\text{min}}$ meeting, the calculation speed index, $T_{\text{seg}}$, must be optimized. A fewer-parameters optimization model is based on the accuracy requirements of $IoU_{\text{min}}$, $I\bar{o}U$, and $T_{\text{seg}}$. The method combines the dense prediction network structure $\eta_{seg}$, backbone network structure $\eta_{main}$, and depth $d_{\text{main}}$. The basic architecture with relatively better $\eta_{seg}$ and $\eta_{main}$ was selected. Improvements in the downsampling of the residual building blocks of the main path and skipped path are proposed to optimize $\eta_{main}$ and $d_{\text{main}}$, and thus obtain a fewer-parameters and optimized atrous convolution architecture network.

Let *k* be the number of classes of the object to be detected with machine vision. The semantic segmentation model requires the identification of *k* + 1 labels, where $L = \{l_0, l_1 \ldots l_k\}$, including the background. Let the number of $l_i$ type of the pixels recognized as the $l_j$ class be $p_{ij}$ (thus, the number of $l_i$ type of the pixels recognized as the $l_i$ class is $p_{ii}$). Then, *PA* and mean $I\bar{o}U$ can be calculated as follows:

$$PA = \frac{\sum_{i=0}^{k} p_{ii}}{\sum_{i=0}^{k} \sum_{j=0}^{k} p_{ij}}, \tag{1}$$

$$I\bar{o}U = \sum_{i=0}^{k} \frac{p_{ii}}{\sum_{j=0}^{k} p_{ij} + \sum_{j=0}^{k} p_{ji} - p_{ii}}. \tag{2}$$

### Fewer-parameters optimization modeling of atrous convolution architecture and selection of basic architecture with $\eta_{seg}$ and $\eta_{main}$ ResNet networks

The accuracy of pixel classification and the amount of segmentation can be measured by using $I\bar{o}U$ and $T_{\text{seg}}$, respectively. The larger the value of $I\bar{o}U$, the higher is the accuracy of pixel

classification. The larger the value of $T_{\text{seg}}$, the larger is the amount of segmentation. In the fewer-parameters optimization of the semantic segmentation network, the aim of fast semantic segmentation technology is to reduce $T_{\text{seg}}$ under the condition that $\bar{IoU}$ meets the requirements of $IoU_{\text{min}}$.

We can build an atrous convolutional neural network by determining backbone network structure $\eta_{\text{main}}$, backbone network depth $d_{\text{main}}$, and dense prediction network structure $\eta_{\text{seg}}$. Therefore, the $\bar{IoU}$ and $T_{\text{seg}}$ of this network can be expressed as $\bar{IoU}(\eta_{\text{seg}}, \eta_{\text{main}}, d_{\text{main}})$ and $T_{\text{seg}}(\eta_{\text{seg}}, \eta_{\text{main}}, d_{\text{main}})$, respectively. The mathematical model of the fewer-parameters optimization based on atrous convolution architecture network can be derived as follows:

$$\begin{cases} \min \ T_{\text{seg}}(\eta_{\text{seg}}, \eta_{\text{main}}, d_{\text{main}}) \\ \text{s.t. } \bar{IoU}(\eta_{\text{seg}}, \eta_{\text{main}}, d_{\text{main}}) \geq IoU_{\text{min}} \end{cases}, \tag{3}$$

where $\eta_{\text{seg}}$, $\eta_{\text{main}}$, and $d_{\text{main}}$ are the optimization parameters.

To optimize $\eta_{\text{seg}}$, $\eta_{\text{main}}$ and $d_{\text{main}}$ in Eq (3), they are first combined. In addition, a basic architecture with relatively better $\eta_{\text{seg}}$ and $\eta_{\text{main}}$ is selected by comparing the values of $\bar{IoU}$ and $T_{\text{seg}}$ of all networks. In this study, the authors used commonly used semantic segmentation networks, such as FCN [16], improved FCN [24], PSPNet [18], and DeepLabv3 [22], available from the representative computer vision deep learning software package, Amazon GluonCV [28]. The $\bar{IoU}$ was obtained from the typical semantic segmentation dataset, PASCAL VOC [1]. The images were segmented using the GeForce GTX 1080Ti GPU hardware environment at $T_{\text{seg}}$ to obtain segmentation results with the same resolution of $1280 \times 1024$. Table 1 lists the performance indexes of the main semantic segmentation network [28], where the relatively better indexes are represented in squares.

The following points should be noted:

1. Among networks 1, 2, 3, and 4 in Table 1, with the same $\eta_{\text{seg}}$, the $\bar{IoU}$ values for $\eta_{\text{main}} =$ ResNet is better than the one for $\eta_{\text{main}} =$ VGG.

2. $\eta_{\text{main}}$ and $d_{\text{main}}$ are the same in networks 2, 3, and 4. The $\bar{IoU}$ for $\eta_{\text{seg}} =$ ASSP is better than those for $\eta_{\text{seg}} =$ PPM or $\eta_{\text{seg}} =$ PSPNet.

3. $\eta_{\text{seg}}$ and $\eta_{\text{main}}$ are the same in networks 4 and 5. The Networks 5 $\bar{IoU}$ shows a slight improvement, and $T_{\text{seg}}$ increases significantly when $d_{\text{main}}$ increases by approximately 50%. Therefore, in this study, the basic architectures of $\eta_{\text{seg}} =$ ASSP and $\eta_{\text{main}} =$ ResNet were selected.

**Table 1. Parameters and indicators of the main semantic segmentation CNN.**

| Number | CNNs | Dense prediction network $\eta_{\text{seg}}$ | Backbone $\eta_{\text{main}}$ | Depth $d_{\text{main}}$ | PASCAL VOC $\bar{IoU}$ | execution time $T_{\text{seg}}$ (ms) |
|---|---|---|---|---|---|---|
| 1 | FCN | FCN | VGG | 16 | 62.2% | 347 |
| 2 | Modified FCN | FCN | ResNet | 101 | 83.6% | 404 |
| 3 | PSPNet | PPM | ResNet | 101 | 85.1% | 510 |
| 4 | DeepLabv3 | ASPP | ResNet | 101 | 86.2% | 495 |
| 5 | DeepLabv3 | ASPP | ResNet | 152 | 86.7% | 646 |

## Improvement of ResNet based on downsampling residual building blocks and fewer-parameters mechanism of $d_{\mathrm{mian}}$

Fig 7 shows the parameter and index of the atrous convolution architecture network for semantic segmentation. The figure also shows the $\eta_{\mathrm{main}}$ hidden-layer type, building block structure, and layer operation parameter of ResNet. $d_{\mathrm{main}}$ represents the number of hidden layers. Fig 8 shows the ResNet model and its bottleneck building block structure. The ResNet backbone network is composed of a $7 \times 7$ convolutional layer, $3 \times 3$ maximum pooling layer, multiple residual building blocks, and downsampled residual building block. Each stage (i.e., the convolutional layer, pooling layer, and downsampling residual building block) reduces the resolution of the feature map by 1/2. The main and skipped paths of the residual and downsampled residual building blocks are connected in series with at most three hidden layers (depth = 3).

The residual unit in the neural network is composed of multiple hidden layers and skip connections. As shown in Fig 9, the input is an $n$-dimensional column vector $\boldsymbol{X}$; the $i$-th hidden layer has a $n_c^i \times n_c^{i-1}$ matrix, $n_c^0$ with weights $\boldsymbol{W}^i$. The output of the $n_c^i$-dimensional column vector $\boldsymbol{H}^i$. Moreover, the residual unit has three hidden layers ($i = 3$), outputs of the $n_c^3$-dimensional column vector, $\boldsymbol{Y}$, are represented as follows [15]:

$$\boldsymbol{Y} = f_{\mathrm{Net}}[\boldsymbol{X}, (\boldsymbol{W}^1, \boldsymbol{W}^2, \boldsymbol{W}^3)] + \boldsymbol{X}. \tag{4}$$

Suppose the actual value is $\boldsymbol{Y}_{\mathrm{GT}}$. The fitting target of the three hidden layers in the residual unit is the residual value between $\boldsymbol{Y}_{\mathrm{GT}}$ and $\boldsymbol{X}$:

$$f_{\mathrm{Net}}[\boldsymbol{X}, (\boldsymbol{W}^1, \boldsymbol{W}^2, \boldsymbol{W}^3)] = \boldsymbol{Y}_{\mathrm{GT}} - \boldsymbol{X}. \tag{5}$$

However, (4) and (5) are satisfied only when the $\boldsymbol{Y}_{\mathrm{GT}}$ and $\boldsymbol{X}$ dimensions are consistent ($n_c^3 = n_c^0$). When $n_c^3 \neq n_c^0$, the $\boldsymbol{X}$ dimension must be transformed to be consistent with the $\boldsymbol{Y}_{\mathrm{GT}}$ dimension. The hidden layer with the $\boldsymbol{W}_{\mathrm{skip}}$ as the $n_c^3 \times n_c^0$ matrix can be added to the skip path to make $\boldsymbol{W}_{\mathrm{skip}}\boldsymbol{X}$ as the $n_c^3$-dimensional column vector. The residual unit operation is as

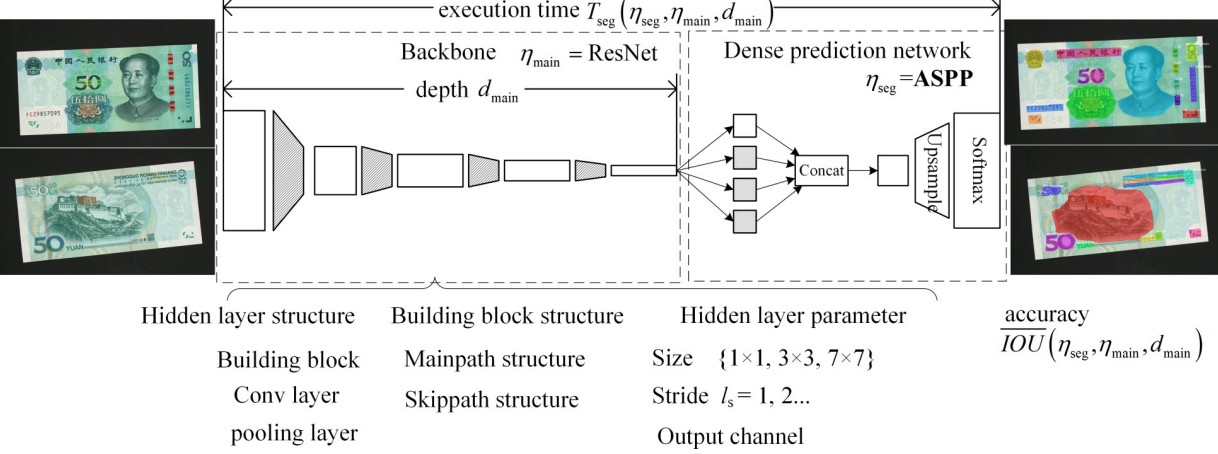

**Fig 7. Parameter and index of atrous convolution architecture network for semantic segmentation.**

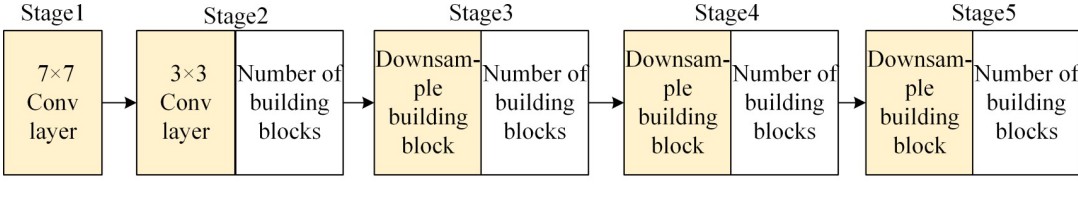

a) ResNet model structure

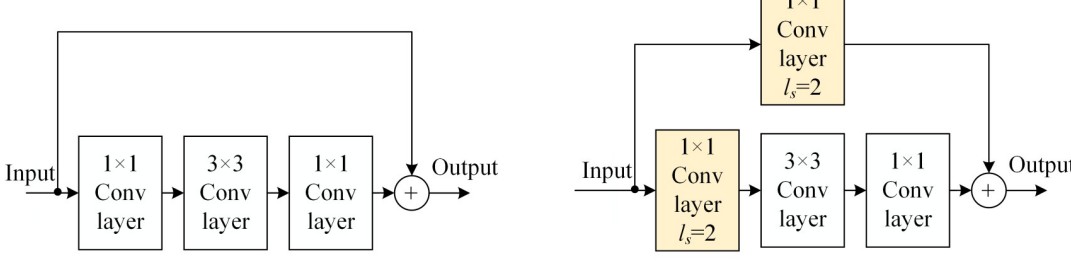

b) Bottleneck block structure    c) Downsample building block structure

**Fig 8. ResNet model and bottleneck building block structure.** a) ResNet model structure, b) bottleneck block structure, and c) downsampled building block structure.

follows:

$$Y = f_{\text{Net}}[X, (W^1, W^2, W^3)] + W_{\text{skip}}X$$
$$\Rightarrow Y = f_{\text{Net}}[X, (W^1, W^2, W^3)] + f_{\text{skip}}(X, W_{\text{skip}}) \tag{6}$$

ResNet downsampling residual building blocks [15] satisfy (6).

The output of the ResNet downsampling residual building block, $f_{\text{Net}}[X,(W^1,W^2,W^3)]$, differs from the $X$ dimension. Output $Y$ and input $X$ represent feature maps, the numbers and sizes of which vary. In the process of changing the number and size of feature maps, if the coefficient of the pixels of the feature map constantly equals 0, the image features are not used; this reduces the accuracy of feature extraction.

Note that 75% coefficients in $W^1$, $W_{\text{skip}}$ of $f_{\text{Net}}(X), f_{\text{skip}}(X)$ are 0. The image-feature utilization is only 25% for input $X$.

Once the downsampling residual building block is optimized, the proportion of input-image features involved in the calculation is improved as much as possible during the change

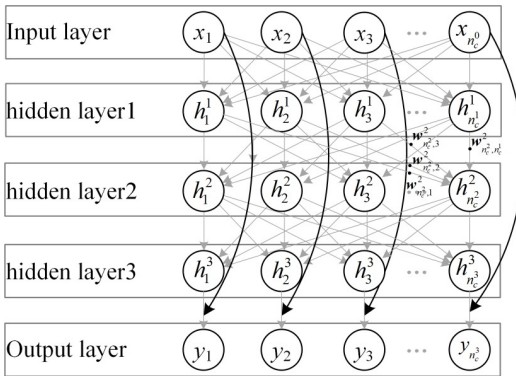

**Fig 9. Residual learning function of ResNet bottleneck building block.**

of the number and size of feature maps; this helps to improve the accuracy of feature extraction. Thus, the requirement of $IoU_{\min}$ is satisfied, and $d_{\text{main}}$ and $T_{\text{seg}}$ are reduced, achieving the improvement of ResNet based on downsampling residual building blocks and fewer-parameters optimization of $d_{\text{main}}$. The fewer-parameters optimization of the atrous convolution architecture network was then performed (which is the contribution of this article).

## Improved ResNet structure based on downsampled residual building blocks

Based on the hidden-layer unit function and image-feature utilization listed in Table 2, the reasonable use of these hidden-layer units can help construct optimized ResNet downsampled residual building blocks.

1) Improved structure of the main path with the 3 × 3 convolutional layer when $l_s = 2$.

Fig 10 shows the modified structure of the downsampled building block with stride 2, 3 × 3 convolutional layers. In the improved scheme, the skip path does not change while the main path changes. The $l_s = 2$ downsampling is moved from the 1 × 1 convolutional layer to the 3 × 3 convolutional layers. Table 2 shows that the dimension of output $f_{\text{Net}}(\boldsymbol{X})$ of the main path does not change in the improved structure. Moreover, the image-feature-utilization rate of the main path increases from the original 25% × 100% × 100% to 100% × 100% × 100%.

2) Improved structure of the skip path with 2 × 2 pooling layer and $l_s = 2$

Fig 11 shows the downsampled building block with the modofoed structure of the skip path of 2 × 2 pooling layers. In the improved scheme, the main path does not change, while the skip path changes with the addition of a 2 × 2 pooling layer. The $l_s = 2$ downsampling moves from 1 × 1 convolution layer to the added 2 × 2 pooling layer. Table 2 shows that the dimension of the skip-path output, $f_{\text{skip}}(\boldsymbol{X})$, does not change in the improved structure. The rate of the image-feature utilization of the skip path increased from the original 25% to 100% × 100%.

3) The improved structures of the main path with the 3 × 3 convolution layer and the skip path with 2 × 2 pooling layer $l_s = 2$.

Fig 12 shows the downsampled building blocks of the main path with stride-2 of 3 × 3 convolutional layers and the modified structure of the skipped path with 2 × 2 pooling layers. The improvement scheme combines the improved structures of Figs 10 and 11. Table 2 shows that the dimensions of the outputs of both the main path [$f_{\text{Net}}(\boldsymbol{X})$] and skip path [$f_{\text{skip}}(\boldsymbol{X})$] do not change in the improved structure. The rate of the image-feature utilization of the main path increases from the original 25% × 100% × 100% to 100% × 100% × 100%. The rate of the image-feature utilization of the skip path increases from the original 25% to 100% × 100%. Figs 10(B), 11(A), 11(B), 12(A) and 12(B) show the improved ResNet structure based on downsampled residual building blocks proposed in this paper.

## Experiments and applications

### Improved model on PASCAL VOC segmentation task

We replace the backbone of DeepLabv3 with the five types of improved structure to obtain the DeepLabv3 improved model. Evaluate the DeepLabv3 improved model with its $\bar{IoU}$ of the

**Table 2. Function and image-feature utilization of general hidden layer structure.**

| Hidden layer structure | Stride $l_s$ | Convolutional | Downsample | Change channel number | Image-feature utilization |
|---|---|---|---|---|---|
| 1×1 Convolutional layer | $l_s = 1$ | —— | —— | ○ | 100% |
| | $l_s = 2$ | —— | ○ | ○ | 25% |
| 3×3 Convolutional layer | $l_s = 1$ | ○ | —— | ○ | 100% |
| | $l_s = 2$ | ○ | ○ | ○ | 100% |
| Pooling layer | $l_s = 2$ | —— | ○ | —— | 100% |

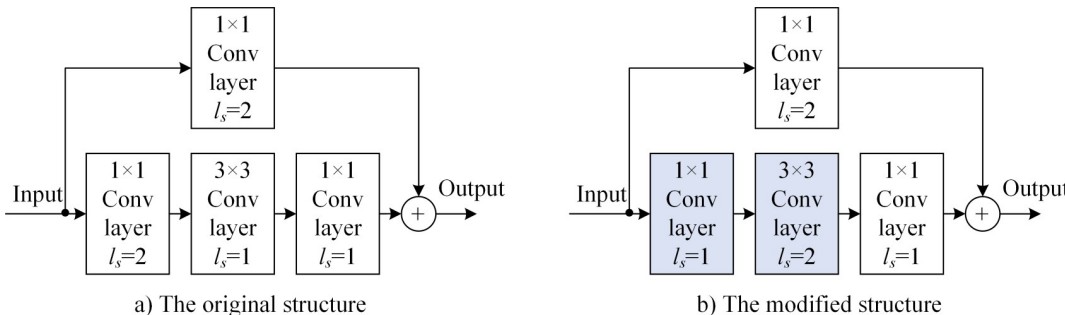

**Fig 10.** Modified structure of the main path with the downsampled building block and stride 2 of 3×3 convolutional layers: a) Original structure and b) modified structure.

PASCAL VOC dataset [1] segmentation task. In the PASCAL VOC segmentation task, $IoU_{min}$ = 85%. We verified the accuracy of the $I\bar{o}U$ index. Segmentation time $T_{seg}$ was obtained for a 1280 × 1024 image, and the comparison of the $I\bar{o}U$ and $T_{seg}$ of the network model of the atrous convolution architecture before and after improving is presented in Table 3.

From Table 3, we can draw the following conclusions:

1) Replacing the backbone of DeepLabv3 with the five types of improved structure proposed in this paper, the accuracy of DeepLabv3 is improved under the same $d_{main}$ situation. For example, depth $d_{main}$ = 50, and each improved structure shows increase in the $I\bar{o}U$ by 85.0%, 85.4%, 85.4%, 86.6%, and 86.5%. However, the values of $T_{seg}$ also show sight increases from 285 to 288, 292, 299, 296, and 303 ms. This shows that improving the downsampled residual building block structure based on the improvement in the utilization of image features is effective for increasing $I\bar{o}U$, and the adoption of the modified structure shown in Fig 12(A) is relatively better for improving the model structure.

2) The improvement in the downsampled residual building block structure produces an increase in $I\bar{o}U$, while reducing $d_{main}$ and $T_{seg}$. For the same $IoU_{min}$, after the improved downsampled residual building block structure is used, $T_{seg}$ is reduced, i.e., to achieve fewer-parameters optimization of the network model relying on the atrous convolution architecture. For example, in Table 3, when ResNet + ASPP is used, with $d_{main}$ = 101, we obtained $I\bar{o}U$ = 86.2%, $T_{seg}$ = 495 ms. In additio, when the structure in Fig 12(A) is used to improve the structure with ASPP, for $d_{main}$ = 50, we obtained $I\bar{o}U$ = 86. 6% and $T_{seg}$ = 296 ms. Thus, we observed improvements in $I\bar{o}U$ and $T_{seg}$ indicators, and the fewer-parameters optimization effect depending on the network model of the atrous convolution architecture is evident. For

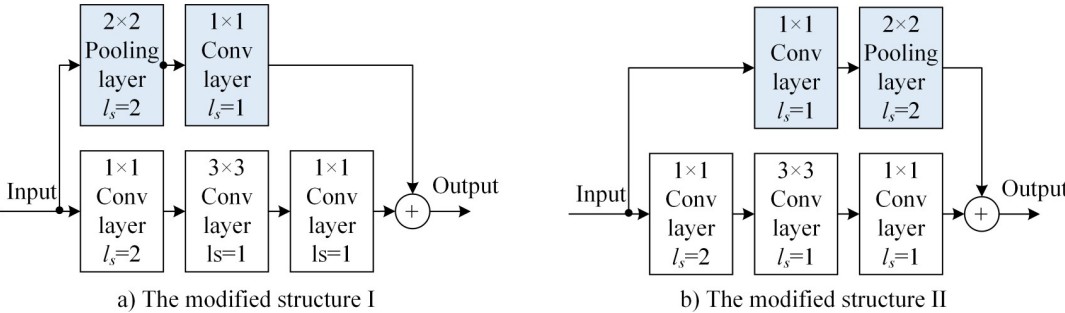

**Fig 11.** The downsampled building block with the modified structure of the 2 × 2-pooling-layer skip path: a) modified structure I and b) modified structure II.

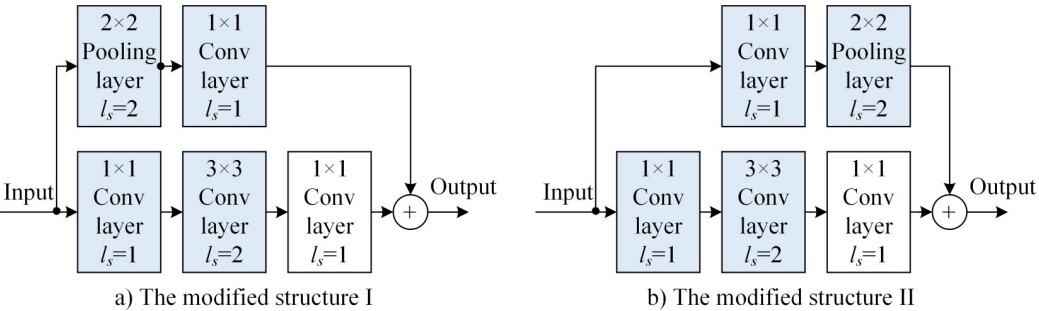

**Fig 12.** Downsampled building block of the main path with stride-2 3 × 3 convolutional layers and modified skip path with 2 × 2 pooling layers: a) modified structure I and b) modified structure II.

example, when ResNet + ASPP is used, for $d_{main}$ = 152, we obtained $I\bar{o}U$ = 86.7% and $T_{seg}$ = 719 ms. Moreover, when using the improved structure with ASPP (Fig 12(A)), for $d_{main}$ = 50, $I\bar{o}U$ = 86.7% but $T_{seg}$ was reduced to 58.8%.

Fig 13 shows semantic segmentation result of the modified DeepLabv3 in the PASCAL VOC task.

### Improved model on CITYSCAPES pixel-level semantic labeling task

We trained each DeepLabv3 (with ResNet50 or modified ResNet50) on the pixel-level semantic labeling task of the CITYSCAPES dataset [30]. To assess performance, CITYSCAPES rely on the PASCAL VOC intersection-over-union metric IoU. Owing to the two semantic granularities, i.e. classes and categories, CITYSCAPES report two separate mean performance scores: $IoU_{class}$ and $IoU_{category}$. In either case, pixels labeled as void do not contribute to the score. We also report the segmentation time of each network run on a GeForce GTX 1080Ti GPU and an Intel i7-5960X CPU. Table 4 presents the performances of different methods on the CITYSCAPES pixel-level semantic labeling task. Table 5 presents the individual classes $IoU$ of different methods in the CITYSCAPES pixel-level semantic labeling task. Fig 14 shows some CITYSCAPES pixel-level semantic labeling results obtained with the DeepLabv3 with different backbone.

From Tables 4 and 5, we can draw the following conclusions:

1) The modified ResNet effectively improved DeepLabv3 performance in CITYSCAPES pixel-level semantic labeling task. Replacing the DeepLabv3's ResNet50 backbone with our

**Table 3. IoU and segmentation time of DeepLab v3 with a modified ResNet in PASCAL VOC segmentation task.**

| | Method | Backbone $\eta_{main}$ | Depth $d_{main}$ = 50 | | Depth $d_{main}$ = 101 | | Depth $d_{main}$ = 152 | |
|---|---|---|---|---|---|---|---|---|
| | | | $I\bar{o}U$ | Tseg | $I\bar{o}U$ | $T_{seg}$ | $I\bar{o}U$ | $T_{seg}$ |
| 1 | DeepLab v3 [22] | ResNet [15] | 84.7% | 285 ms | 86.2% | 495 ms | 86.7% | 719 ms |
| 2 | DeepLab v3* | ResNet v2 [29] | 84.4% | 276 ms | 85.9% | 469 ms | 86.7% | 677 ms |
| 3 | DeepLab v3* | stride 2 3×3 conv layers mainpath modified structure | 85.0% | 288 ms | 86.7% | 474 ms | 87.2% | 680 ms |
| 4 | DeepLab v3* | 2×2 pooling layers skippath modified structure I | 85.4% | 292 ms | 87.1% | 492 ms | 87.6% | 691 ms |
| 5 | DeepLab v3* | 2×2 pooling layers skippath modified structure II | 85.4% | 299 ms | 87.0% | 502 ms | 87.6% | 704 ms |
| 6 | DeepLab v3* | stride 2 3×3 conv layers mainpath and 2×2 pooling layers skippath modified structure I | **86.6%** | **296 ms** | 88.1% | 513 ms | 88.2% | 698 ms |
| 7 | DeepLab v3* | stride 2 3×3 conv layers mainpath and 2×2 pooling layers skippath modified structure II | 86.5% | 303 ms | 88.1% | 523 ms | 88.2% | 711 ms |

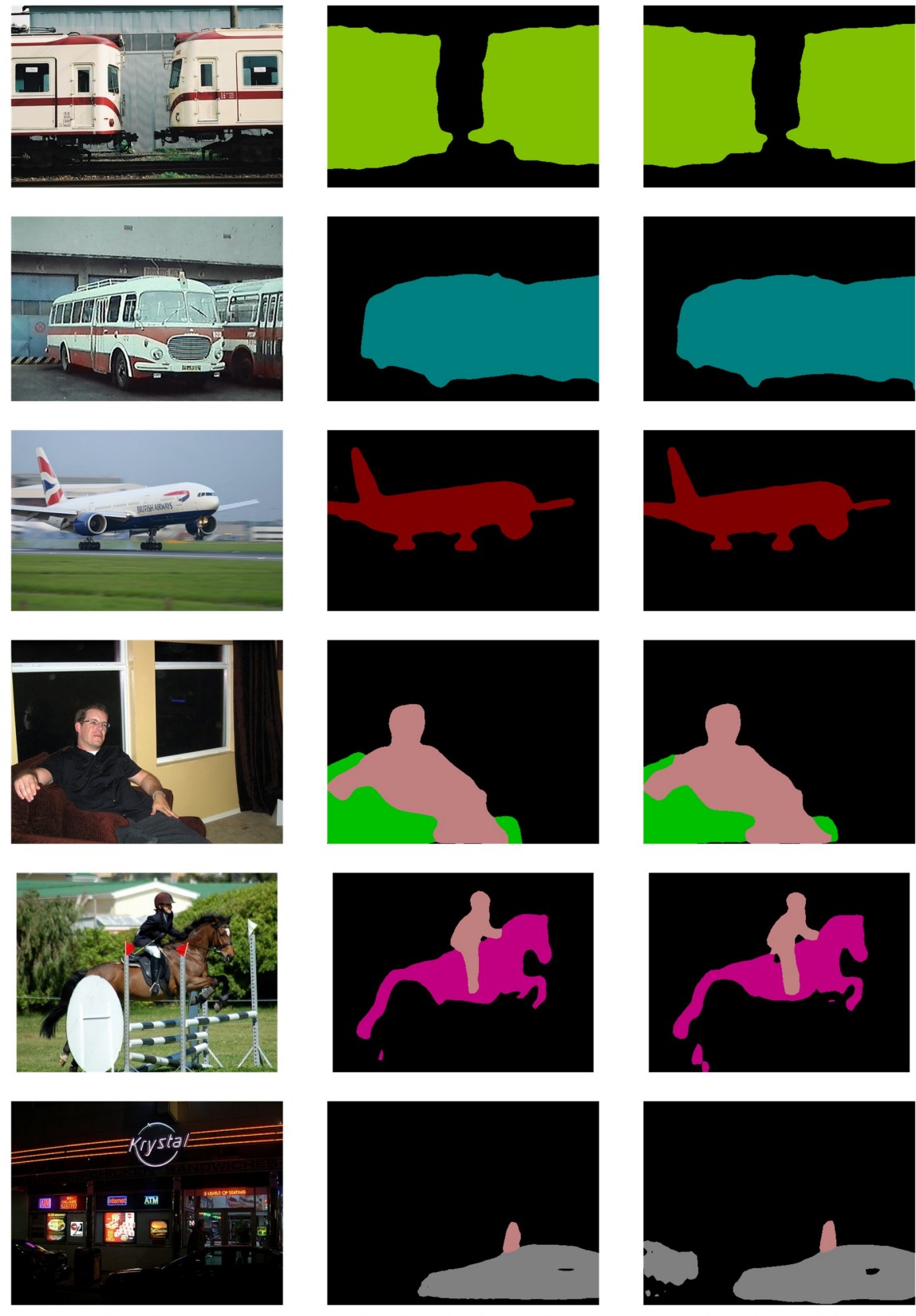

Oringinal image          DeepLab v3 with Resnet50          DeepLab v3 with modified Resnet50

**Fig 13. PASCAL VOC semantic segmentation by modified DeepLabv3.**

**Table 4. Performances of different methods on the CITYSCAPES pixel-level semantic labeling task.**

| Method | $IoU_{class}$(%) | $IoU_{category}$(%) | Segmentation time $T_{seg}$(ms) |
|---|---|---|---|
| DeepLab [19] | 64.5 | 83.7 | —— |
| DeepLab v2 [21] | 70.4 | 86.4 | —— |
| DeepLab v3 with ResNet50 [22] | 78.7 | 89.8 | **494ms** |
| DeepLab v3 with modified ResNet50 [ours] | **79.9** | **91.6** | 497ms |

modified ResNet50 (stride 2 3×3 conv layers mainpath and 2×2 pooling layers skippath modified structure I), the $IoU_{class}$ is improved from 78.7% to 79.9%, and the $IoU_{category}$ is improved from 89.8% to 91.6%. The segmentation time $T_{seg}$ increased slightly from 494ms to 497 ms.

2) The modified ResNet also improved most of the individual classes IoU of the DeepLabv3. Especially, the wall, pole, traffic light, and rider IoU have increased by more than 5%. The traffic sign, person, train, and motor cycle IoU have increased by more than 2%.

## Improved model on machine vision banknote anti-counterfeit segmentation

The following experiment was conducted using the 2019 version of the CNY machine vision segmentation and recognition application. In this task, the quantitative value for $IoU_{min}$ was 90%. The experimental model includes an improved ResNet structure before and after the atrous convolution architecture network. In the DeepLab v3 semantic segmentation model, the dense prediction network was set as ASPP, the backbone network was replaced with various ResNet structures, the entire semantic segmentation network was trained with the 2019 version of the CNY image dataset. We verified the accuracy of $\eta_{seg}$ and $T_{seg}$.

1) The machine vision system adopts an industrial camera, MV-CA013-10GC, with an MVL-HF2528M-6MP industrial lens, a bar light source with FOV = 18.33˚, and an imaging resolution of 1280 × 1024.

2) For the image-dataset construction, the object plane is perpendicular to the optical axis. The object distance is 400 mm and various denominations, with 25 pieces of front and back renminbi, were collected with various angle. The 2019 version of the CNY image dataset (200 image) comprises the serial numbers on the CNY, magnetic optical variable ink (OVMI), security line denomination numbers, five patterns of visual features, etc..

3) Hardware condition: The hardware is the same as that used in the aforementioned experiment: GeForce GTX 1080Ti GPU.

Table 6 presents the $I\bar{o}U$, execution time, and $AP^{IoU_T=0.1}$ of DeepLab v3 with a modified ResNet in the segmentation task of CNY anti-counterfeit features. Fig 15 shows CNY anti-counterfeit feature segmentation with diffuse reflection machine vision and modified DeepLab v3.

From Table 6, we can draw the following conclusions: The modified ResNet effectively improved DeepLabv3 performance in CNY anti-counterfeit feature-segmentation task.

**Table 5. Individual classes IoU of DeepLabv3 with different backbone in CITYSCAPES pixel-level semantic labeling task.**

| Method | road | sidewalk | building | wall | fence | pole | traffic light | traffic sign | vegetation | terrain |
|---|---|---|---|---|---|---|---|---|---|---|
| DeepLab v3 with ResNet50 [22] | 98.6% | 86.4% | 92.8% | 52.4% | 59.7% | 59.6% | 72.5% | 78.3% | 93.3% | **72.8%** |
| DeepLab v3 with modified ResNet50 [ours] | **98.7%** | **87.0%** | **93.5%** | **57.9%** | **60.4%** | **70.9%** | 77.9% | **81.4%** | **93.7%** | 72.8% |
| Method | sky | person | rider | car | truck | bus | train | motor cycle | bicycle | average |
| DeepLab v3 with ResNet50 [22] | 95.5% | 85.4% | 70.0% | 95.7% | **75.4%** | **84.1%** | 75.1% | 68.7% | **75.0%** | 78.7% |
| DeepLab v3 with modified ResNet50 [ours] | **95.6%** | **87.9%** | **75.3%** | **96.1%** | 65.8% | 80.5% | **78.7%** | **72.8%** | 70.8% | **79.9%** |

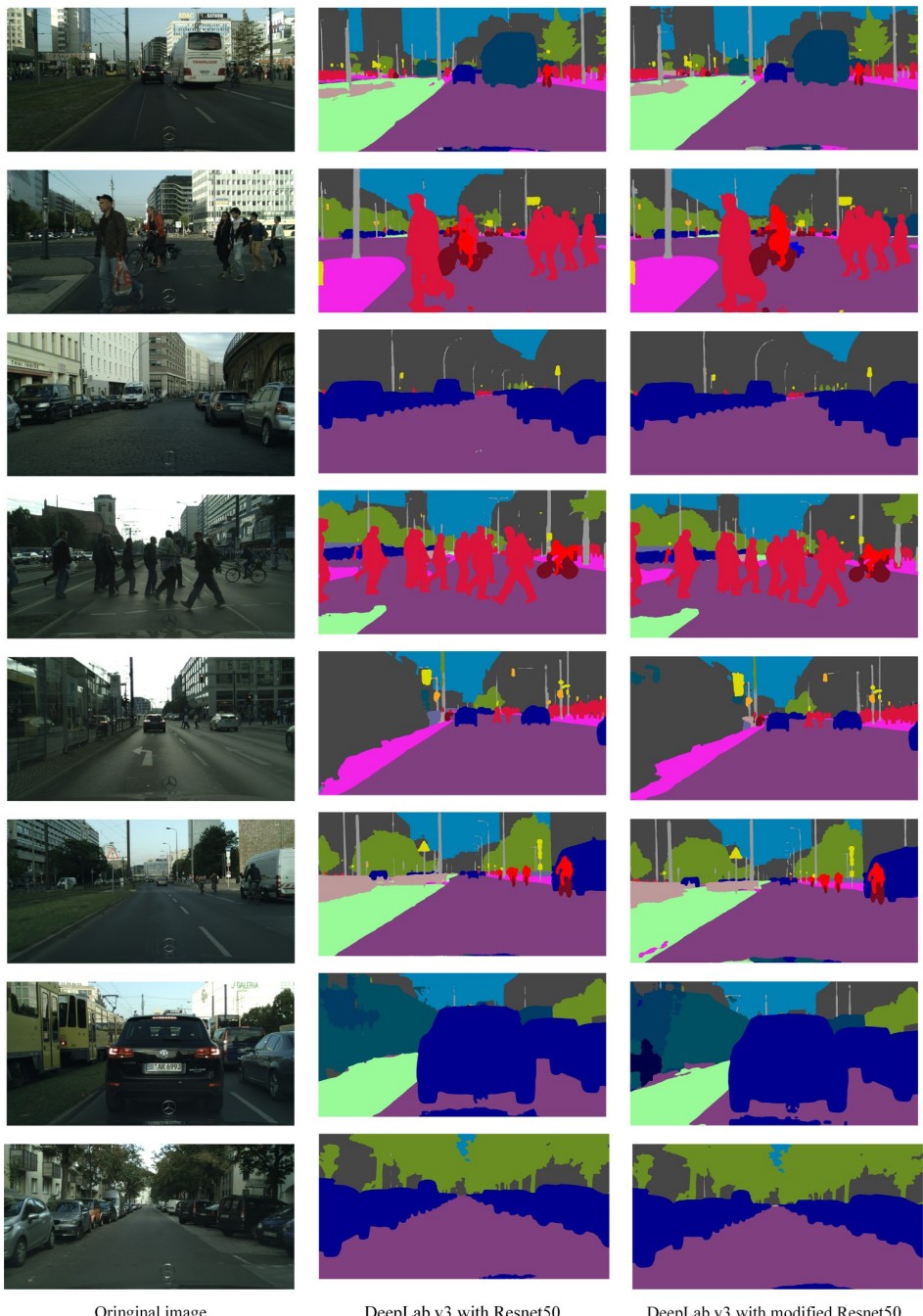

Oringinal image  DeepLab v3 with Resnet50  DeepLab v3 with modified Resnet50

**Fig 14. CITYSCAPES pixel-level semantic labeling by modified DeepLabv3.**

Replacing the DeepLabv3's ResNet backbone with our modified ResNet (stride 2 3×3 conv layers mainpath and 2×2 pooling layers skippath modified structure I), the accuracy $I\bar{o}U$ is improved under the same $d_{main}$ situation. For example, in depth $d_{main}$ = 50, our modified ResNet make $I\bar{o}U$ increase from 90.3% to 91.3%, and $T_{seg}$ show sight increases from 287 ms to 295 ms.

**Table 6. IOU and execution time of Deeplab v3 with a modified ResNet in CNY anti-counterfeit feature-segmentation task.**

| | Method | Backbone $\eta_{\text{main}}$ | Depth $d_{\text{main}} = 50$ | | Depth $d_{\text{main}} = 152$ | |
|---|---|---|---|---|---|---|
| | | | $I\bar{o}U$ | $T_{\text{seg}}$ | $I\bar{o}U$ | $T_{\text{seg}}$ |
| 1 | DeepLab v3 [22] | ResNet [15] | 90.3% | 287 ms | 91.3% | 722 ms |
| 2 | DeepLab v3* | ResNet v2 [29] | 89.9% | 282 ms | 91.1% | 714 ms |
| 3 | DeepLab v3* | stride 2 3×3 conv layers mainpath and 2×2 pooling layers skippath modified structure I | 91.3% | 295 ms | 92.5% | 735 ms |

The conclusions drawn in Table 6 are consistent with the conclusions drawn for the PAS-CAL VOC dataset (Table 3). That is, the improved downsampled residual building block structure based on the improvement of the image-feature-utilization rate is helpful for improving $I\bar{o}U$. Moreover, by adopting the improved structure with relatively better $I\bar{o}U$ and $T_{\text{seg}}$, the fewer-parameters optimization of the network model of the atrous convolution architecture is evident.

## Conclusions

In this study, fewer-parameters optimization was analyzed according to the atrous convolution network architecture model. For this, we selected ASPP and ResNet as the replacements for

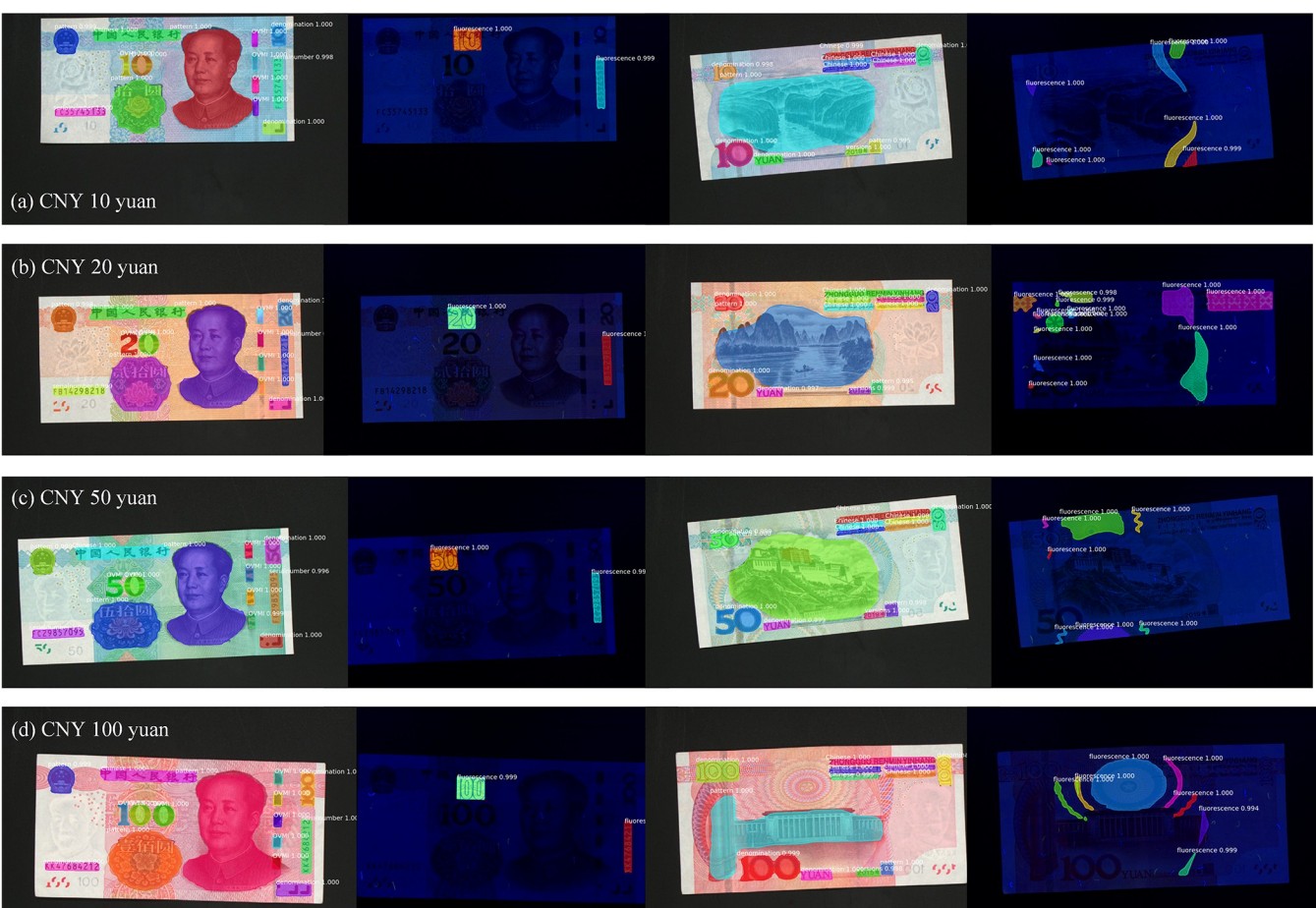

**Fig 15. CNY anti-counterfeit features segmentation by modified DeepLabV3.**

$\eta_{\mathrm{seg}}$ and $\eta_{\mathrm{main}}$ basic architectures, respectively. Our analysis indicates that the optimization of the downsampled residual block increases the number of feature maps and the proportion of input image features used as much as possible. Specifically, in the application of the number and dimension of feature map during the change of the size of the input image, the accuracy of feature extraction is improved.

The proposed method innovatively modified the structures of the main path with a $3 \times 3$ convolution layer, $2 \times 2$ skip path, and $l_s = 2$ pooling layer, with respect to five modified ResNet structures based on the downsampled residual building blocks, and thus improved the utilization of image features. As shown in the simulation experiments, under the same $\eta_{\mathrm{seg}} = $ ASPP, by using the modified ResNet instead of the conventional ResNet, the $d_{\mathrm{main}}$ changed from 152 to 50, enabling $IoU$ to change from 86.7% to 86.6% and $T_{\mathrm{seg}}$ to change from 719 to 296 ms (decreased 58.8%). The fewer-parameters optimization of the atrous convolution architecture network model was also verified in the application of machine-vision segmentation recognition of 2019 version of CNY with an evident fewer-parameters optimization effect.

We have proposed the modified structure of the downsampled residual blocks to improve the utilization of image features that showed evident fewer-parameters optimization effects. The next step is to study the modified structure of more complicated downsampled residual blocks in networks such as ResNext and ResNest.

## Author Contributions

**Conceptualization:** Liu Guixiong.

**Data curation:** Binyuan He.

**Formal analysis:** Jian Huang, Binyuan He.

**Funding acquisition:** Jian Huang, Liu Guixiong.

**Investigation:** Jian Huang.

**Project administration:** Jian Huang, Liu Guixiong.

**Resources:** Liu Guixiong.

**Software:** Jian Huang.

**Supervision:** Liu Guixiong.

**Validation:** Jian Huang, Liu Guixiong.

**Visualization:** Binyuan He.

**Writing – original draft:** Jian Huang.

**Writing – review & editing:** Jian Huang, Liu Guixiong, Binyuan He.

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
