## [Decision Letter · Decision Letter 0]

1 Dec 2020

PONE-D-20-27701

Fast Semantic Segmentation Method for Machine Vision Inspection Based on a Fewer-Parameters Atrous Convolution Neural Network

PLOS ONE

Dear Dr. Guixiong,

Thank you for submitting your manuscript to PLOS ONE. After careful consideration, we feel that it has merit but does not fully meet PLOS ONE’s publication criteria as it currently stands. Therefore, we invite you to submit a revised version of the manuscript that addresses the points raised during the review process.

We look forward to receiving your revised manuscript.

Kind regards,

Jeonghwan Gwak, PhD

Academic Editor

PLOS ONE

Journal Requirements:

2. Please ensure that you refer to Figure 13 in your text as, if accepted, production will need this reference to link the reader to the figure.

Additional Editor Comments:

Please revise this manuscript to fully respond to reviewer 2's comments.

Reviewers' comments:

Reviewer's Responses to Questions

**Comments to the Author**

1. Is the manuscript technically sound, and do the data support the conclusions?

Reviewer #1: Yes

Reviewer #2: Yes

2. Has the statistical analysis been performed appropriately and rigorously? 

Reviewer #1: Yes

Reviewer #2: Yes

3. Have the authors made all data underlying the findings in their manuscript fully available?

Reviewer #1: Yes

Reviewer #2: Yes

4. Is the manuscript presented in an intelligible fashion and written in standard English?

Reviewer #1: Yes

Reviewer #2: Yes

5. Review Comments to the Author

Reviewer #1: In this paper, the authors present fast semantic segmentation method for machine vision inspection based on a fewer parameters atrous convolution neural network. The article is well constructed, the experiments were well conducted, and analysis was well performed.

Reviewer #2: This work designed a fewer-parameters optimization mathematical model according to the atrous convolution architecture network to decrease segmentation time while maintaining the detection accuracy, although there are some significant improvements that the authors reflected previous reviewers, there are several minor aspects the authors should consider:

1. More recent works should be included, both CNN-based semantic segmentation and Atrous CNN-based works to enlighten your novelty in the related works section.

2. The authors merely presented their observations in the results section. However, to make the experimental results more convinced, please give more explanations on the results you obtained.

3. Please increase the figures' resolution if possible.

6. PLOS authors have the option to publish the peer review history of their article (what does this mean?). If published, this will include your full peer review and any attached files.

Reviewer #1: No

Reviewer #2: No

---

## [Author Response · Author response to Decision Letter 0]

15 Dec 2020

Respones to Reviewers

Dear Editor,

This article have sent to Plos One as Semantic Segmentation Visual Detection Technology Based on Downsampling Porous Convolution Architecture Network Model Lightweight Optimization (ONE-D-20-15502), and receive a decision letter. 

Thank you very much for your time and discussion on this manuscript. We also thank the Reviewers very much for their constructive comments. We have revised the manuscript accordingly. The replies are follows:

Journal Requirements:

Reply: According to the editor’s suggestion, we edit our manuscript to meets PLOS ONE's style requirements, including those for file naming.

2. Please ensure that you refer to Figure 13 in your text as, if accepted, production will need this reference to link the reader to the figure.

Reply: According to the editor’s suggestion, we refer to Figure 13 in our text. 

Additional Editor Comments: Please revise this manuscript to fully respond to reviewer 2's comments.

Reply: According to the editor’s suggestion, we revise this manuscript to fully respond to reviewer 2's comments.

Reviewer #1:

 In this paper, the authors present fast semantic segmentation method for machine vision inspection based on a fewer parameters atrous convolution neural network. The article is well constructed, the experiments were well conducted, and analysis was well performed.

Reviewer #2:

This work designed a fewer-parameters optimization mathematical model according to the atrous convolution architecture network to decrease segmentation time while maintaining the detection accuracy, although there are some significant improvements that the authors reflected previous reviewers, there are several minor aspects the authors should consider:

1. More recent works should be included, both CNN-based semantic segmentation and Atrous CNN-based works to enlighten your novelty in the related works section.

Reply: According to the reviewer’s suggestion, we list more recent works in the related works section, both CNN-based semantic segmentation and Atrous CNN-based works that enlighten my novelty.

2. The authors merely presented their observations in the results section. However, to make the experimental results more convinced, please give more explanations on the results you obtained.

Reply: According to the reviewer’s suggestion, we give more explanations on the results that we obtained. Especially the CITYSCAPES pixel-level semantic labeling results and banknote anti-counterfeit segmentation results

3. Please increase the figures' resolution if possible.

Reply: According to the reviewer’s suggestion, we increase the figures' resolution to 600dpi, which is twice as high as the PLOS ONE requirements.

While revising your submission, please upload your figure files to the Preflight Analysis and Conversion Engine (PACE) digital diagnostic tool

Reply: As Request, We upload our figure files to the PACE digital diagnostic tool, and upload the PACE output figure files to PLOSONE.

---

## [Decision Letter · Decision Letter 1]

14 Jan 2021

Fast Semantic Segmentation Method for Machine Vision Inspection Based on a Fewer-Parameters Atrous Convolution Neural Network

PONE-D-20-27701R1

Dear Dr. Guixiong,

We’re pleased to inform you that your manuscript has been judged scientifically suitable for publication and will be formally accepted for publication once it meets all outstanding technical requirements.

Kind regards,

Jeonghwan Gwak, PhD

Academic Editor

PLOS ONE

Additional Editor Comments (optional):

Reviewers' comments:

Reviewer's Responses to Questions

**Comments to the Author**

1. If the authors have adequately addressed your comments raised in a previous round of review and you feel that this manuscript is now acceptable for publication, you may indicate that here to bypass the “Comments to the Author” section, enter your conflict of interest statement in the “Confidential to Editor” section, and submit your "Accept" recommendation.

Reviewer #1: All comments have been addressed

Reviewer #2: All comments have been addressed

2. Is the manuscript technically sound, and do the data support the conclusions?

Reviewer #1: Yes

Reviewer #2: Yes

3. Has the statistical analysis been performed appropriately and rigorously? 

Reviewer #1: Yes

Reviewer #2: Yes

4. Have the authors made all data underlying the findings in their manuscript fully available?

Reviewer #1: Yes

Reviewer #2: Yes

5. Is the manuscript presented in an intelligible fashion and written in standard English?

Reviewer #1: Yes

Reviewer #2: Yes

6. Review Comments to the Author

Reviewer #1: The authors reflected all of the reviewers' comments. Especially, the authors improved their manuscript by adding more recent works in the related works section, and more explanations on the results. Overall, the manuscript is well constructed, the experiments were well conducted, and analysis was well performed. Also, I find the manuscript very clearly written. For these reasons, I am strongly inclined to recommend it for publication.

Reviewer #2: After reflecting previous reviewers' comments, some significant improvements have been made. In overall, the organization of the paper is well-constructed, the experiments are well-conducted, the results are well-analyzed and the manuscript is well-written.

7. PLOS authors have the option to publish the peer review history of their article (what does this mean?). If published, this will include your full peer review and any attached files.

Reviewer #1: No

Reviewer #2: No

---

## [Editor Report · Acceptance letter]

28 Jan 2021

PONE-D-20-27701R1 

Fast Semantic Segmentation Method for Machine Vision Inspection Based on a Fewer-Parameters Atrous Convolution Neural Network 

Dear Dr. Guixiong:

I'm pleased to inform you that your manuscript has been deemed suitable for publication in PLOS ONE. Congratulations! Your manuscript is now with our production department. 

Kind regards, 

on behalf of

Dr. Jeonghwan Gwak 

Academic Editor

PLOS ONE